# The First 1000 Days: Impact of Prenatal Tobacco Smoke Exposure on Hospitalization Due to Preschool Wheezing

**DOI:** 10.3390/healthcare9081089

**Published:** 2021-08-23

**Authors:** Cyrielle Collet, Michael Fayon, Florence Francis, François Galode, Stephanie Bui, Stephane Debelleix

**Affiliations:** 1CHU de Bordeaux, Département de Pédiatrie, Unité de Pneumologie Pédiatrique, Centre d’Investigation Clinique (CIC 1401), Place Amélie Raba-Léon, F-33076 Bordeaux, France; cys.collet@gmail.com (C.C.); francois.galode@chu-bordeaux.fr (F.G.); stephanie.bui@chu-bordeaux.fr (S.B.); stephane.debelleix@chu-bordeaux.fr (S.D.); 2Université de Bordeaux, Centre de Recherche Cardio-Thoracique de Bordeaux, Inserm U1045, F-33000 Bordeaux, France; 3Centre INSERM U1219-Bordeaux Population Health France, Institut de Santé Publique, d’Epidemiologie et de Développement (ISPED), Univ. Bordeaux, F-33000 Bordeaux, France; florence.francis@u-bordeaux.fr

**Keywords:** asthma, child, preschool, prenatal exposure, delayed effects, pregnancy, smoking, adverse effect

## Abstract

Preschool wheezing and related hospitalization rates are increasing. Prenatal tobacco smoke exposure (PTSE) increases the risk of wheezing, yet >20% of French women smoke during pregnancy. In this observational retrospective monocentric study, we assessed the link between PTSE and hospital admissions. We included infants <2 years of age admitted for acute wheezing. A phone interview with mothers was completed by electronic records. The primary endpoint was the ratio of cumulative duration of the hospitalization stays (days)/age (months). 129 children were included (36.4% exposed to PTSE vs. 63.6% unexposed). There was a significant difference in the duration of hospitalization/age: 0.9 days/month (exposed) vs. 0.58 days/month (unexposed) (*p* = 0.008). Smoking one cigarette/day during pregnancy was associated with an increase in hospitalization duration of 0.055 days/month (r = 0.238, *p* = 0.006). In the multi-variable analysis, this positive association persisted (β = 0.04, *p* = 0.04; standardized β = 0.27, *p* = 0.03). There was a trend towards a dose-effect relationship between PTSE and other important parameters associated with hospital admissions. We have demonstrated a dose-effect relationship, without a threshold effect, between PTSE and duration of hospitalization for wheezing in non-premature infants during the first 2 years of life. Prevention campaigns for future mothers should be enforced.

## 1. Introduction

It is estimated that 15–37% of women smoke while pregnant [1]. In 2017 in France smoking during pregnancy was estimated to affect between 20% and 25% of pregnant women, and was more frequent amongst the youngest and least educated women [2]. There is a wide range of effects of prenatal tobacco smoke exposure (PTSE) on mortality and morbidity in infants and children. Maternal smoking increases the risk of obstetric complications, preterm birth, low birth weight and poor intrauterine growth, infant death (Sudden Infant Death Syndrome), neurodevelopmental and behavioral problems and possibly childhood cancer [1,3]. Regarding the lung, it is generally accepted that there is reduced and disturbed intra-uterine lung growth and as a consequence an increased incidence of acute lower airway infections (0–3 years), persisting reduced lung function and increased incidence of wheeze (0–5 years) [1,3,4].

Preschool wheeze is a multifactorial disease whose expression depends on the environment. Its prevalence and related hospitalization rate are increasing [5,6]. The influence of PTSE on the development of preschool wheeze is known to cause at least a 20% increase in wheezing and asthma rates. The wheezing phenotype most closely associated with PTSE is transient early wheezing, i.e., at least one lower respiratory tract illness with wheezing during the first three years of life but no wheezing at six years of age. This is accompanied by significantly lower length-adjusted values for V_sub max_FRC in infancy [7].

Most studies describe a pathophysiological and clinical dose-effect relationship, but with varying thresholds [5,6]. Moreover, the clinical impact in mothers and infants regarding hospitalizations during the first 1000 days is not well known. In the present study, we thus analyzed the dose-effect relationship between PTSE and the duration of hospitalization for wheezing in infants under 2 years of age in a French tertiary-care regional reference center, i.e., the Bordeaux Children’s University Hospital. We also analyzed the dose-effect relationship between this exposure and the severity of dyspnea during hospital admissions, how mothers experienced their child’s hospitalizations and the associated rate of parental workplace absenteeism, the number of consultations in pediatric emergency departments’ services for wheezing and the age of the first episode of wheezing.

## 2. Methods

This was a retrospective single-center observational study conducted from July 2016 to February 2017. The Ethics Committee and Institutional Review Board of Bordeaux University Hospital validated this project and waived the necessity of written informed consent. The mothers called were informed about the objective of the study and agreed to answer the questionnaire. Data use met the requirements of the CNIL, which is the French data protection authority, and all data were anonymized. Patients included had been hospitalized at least once for an episode of wheezy dyspnea, from January 2015 to May 2016 at Bordeaux Children’s Hospital. They were less than 2 years old at the time of hospitalization. Patients had had at least three episodes of wheezy dyspnea in their lifetime, or at least two episodes with a family or personal history of atopy. Patients who were born preterm (<37 weeks of gestation) and who had comorbidities such as heart disease, immune deficiency, degenerative neuromuscular pathology, upper and lower airway malformations, or severe laryngo-tracheomalacia were excluded.

The data was collected from the patients’ files (Dx Care^®^ software, Medasys, Paris, France) and completed by a maternal questionnaire. For the latter, we conducted a telephone questionnaire interview with the mothers of hospitalized infants. We collected patient characteristics, the existence of family or personal atopy, the main risk factors for preschool wheezing, API (Asthma Predictive Index) [8] and PIAMA (Prevention and Incidence of Asthma and Mite Allergy) [9] scores to phenotype asthma, and socio-economic level as assessed by educational level (if more or less than grade 12). We also collected data regarding hospitalization and medical respiratory history. The severity of dyspnea during hospitalization was defined by oxygen requirement, and/or complications such as pneumonia or atelectasis. Mothers graded from 0 (very bad experience) to 5 (good) their experience of their child’s hospitalizations. We documented the rate of parental workplace absenteeism and the number of emergency visits for wheezing, as well as PTSE as described by the mothers. The other types of passive smoking were also noted, i.e., as passive smoking for pregnant women, partner smoking and post-natal passive smoking. We noted the mothers’ behaviors related to smoking cessation or reduction and the information given to mothers regarding the risk of asthma for the child if exposed to prenatal tobacco smoke.

The primary endpoint was the duration in days of hospitalization for wheezy dyspnea related to the patient’s age in months, i.e., the ratio “duration of hospitalization/age”. If the patient had been hospitalized several times, the duration of hospitalization was the cumulative duration of all stays normalized to the age at the last hospital admission.

Statistical analysis was performed using several types of software: Excel 2010, Microsoft Redmond, USA; NCSS 2001, Kaysville, Utah, USA and Sas 9.4, SAS Institute Inc., Cary, NC, USA. Data were expressed as means ± SD or as median (IQR). Quantitative and qualitative values were compared using Mann-Whitney and Fisher tests. A linear multivariable regression was performed with Sas 9.4 using Proc reg. Standardized coefficients were calculated, and the same procedure applied.

## 3. Results

A total of 237 patients were recruited. Of the 129 responses obtained by the telephone questionnaire interview, 47/129 (36.4%) infants had been exposed to prenatal tobacco smoking and 82/129 (63.6%) infants had not. The general characteristics of the study population are summarized in Table 1.

Compared to the population of patients with wheezing/asthma admitted during the same time frame, included patients were younger upon their first wheezing episodes and had more severe disease. However, the sex ratio was identical, with a greater proportion of males (Table 2).

Regarding exposure to passive smoking and mothers’ associated behavior, 49/129 (38%) of women had been exposed to passive smoking during their pregnancy (partner smoking), 31/47 (66%) in the exposed group and 18/82 (22%) in the unexposed group. 

Fifty six infants (43%) were exposed to post-natal passive smoking, 41/47 (98%) in the exposed group, 10/82 (12%) in the unexposed group.

In the exposed group, 87% of infants were exposed until the third trimester pregnancy and 10% during the first trimester, only. Six percent of the mothers had quit smoking in anticipation of their pregnancy and 83% had reduced their consumption. Eight and a half percent of women reported measuring exhaled carbon monoxide during pregnancy follow-up, 66% would have liked more help regarding smoking cessation and 60% had received information about the risk of asthma for their child if they were smoking during their pregnancy. The mean reported daily cigarette consumption throughout pregnancy was 7.6 ± 4.7 cigarettes. Forty-nine percent (23/47) of mothers reported smoking fewer than five cigarettes per day, 13/47 (27.6%) between six and 10 cigarettes and 11/47 (23.4%) between 11 and 20 cigarettes. In the group exposed to antenatal smoking (*n* = 47, 36.4%), 48.9% of mothers reported smoking fewer than five cigarettes per day, 27.6% between six and 10 cigarettes and 23.4% between 11 and 20 cigarettes.

Regarding the primary outcome measure (Table 3), there was a significant difference in the hospitalization duration/age ratio between the exposed and unexposed groups (0.9 days/month in the exposed group, 0.58 days/month in the unexposed group, *p* = 0.008). Of note, the median length of hospitalization/age in the overall cohort was 0.6 days/month ranging from 0.1 to 6.5 days/month. Median hospitalization duration/age based on PTSE increased with exposure. There was a difference of 0.2 days/month between the unexposed group and the group with low cigarette consumption ((1–5) cigarettes per day), a difference of 0.35 days/month between the low and medium exposure groups ((6–10) cigarettes), and a difference of 0.32 days/month between the medium and high level exposed groups ((11–20) cigarettes). There was a 54% increase in duration of hospitalization in the exposed versus the unexposed group.

In the univariate analysis, as depicted in Figure 1, there was a linear relationship between prenatal tobacco smoke exposure and duration of hospitalization/age (r = 0.238, slope = 0.055, (95% CI: 0.02–0.09), *p* = 0.006). Smoking one cigarette per day during pregnancy was statistically associated with an increase in hospital stay of 0.055 days/month over the first 2 years of life. 

Upon multivariable analysis, after adjusting for confounding factors related to other risk factors for wheezing (such as the passive exposition for the pregnant women or in the post-natal period, gender, intrauterine growth restriction, familial atopy), smoking one cigarette per day during pregnancy was statistically associated with an increase in hospital stay of 0.06 days/month of life over the first 2 years of life (*p* = 0.03), as depicted in Table 4. The standardized β of this association was 0.27 after standardization of the coefficients, i.e., an increase by one standard deviation of the independent variable (number of cigarettes smoked per day during pregnancy) induced an increase of 0.27 SD of the hospital stay (in days/month).

Regarding the secondary endpoints (Table 5), the descriptive analysis of the relation between the quantification of the exposure and the secondary endpoints revealed a dose-effect trend. Upon multivariable analysis, there was no significant association between PTSE and all the secondary endpoints: the severity of the dyspnea during hospitalization: OR = 1.033, *p* = 0.67, CI_95%_ (0.89–1.19); how mothers experience their child’s hospitalization: β = −0.04 and *p* = 0.32, CI_95%_ (−0.12–0.02); rate of parental work absenteeism: β = 1.52 and *p* = 0.43, CI_95%_ (1.25–1.74); age of first wheezing episode: β = −0.12 and *p* = 0.26, CI_95%_ (−0.33–0.09); number of emergency department visits: β = 0.079 and *p* = 0.25, CI_95%_ (0.01–0.17).

## 4. Discussion

In the present study, we found a significant dose-effect relationship between prenatal tobacco smoke exposure and the duration of hospitalization for wheezing in non-premature infants below 24 months of age. For each cigarette smoked per day during pregnancy, there was a risk of increasing the duration of hospitalization by 0.055 days per month over the infant’s first two years of life. For a 12-month-old infant exposed to 10 cigarettes per day antenatally, there was a risk of increasing his stay by 6.6 days and 4.8 days, according to the univariate analysis and the multivariable analysis, respectively. There was a 54% increase in the duration of hospitalization in the exposed vs. the unexposed group. There was no dose threshold above which the duration of hospitalization would increase further, or below which there would be no risk. Regarding the secondary endpoints, there were non-significant trends towards a similar dose-effect relationship. The severity score tended to increase (20% difference between unexposed group and group exposed to heavy smoking), the mothers’ grading of how they experienced the admission tended to decrease (remaining below 2.5 indicating that hospitalization is difficult for all patients) and the age of the first wheezing episode tended to decrease with exposure. 

### 4.1. Demographics and Risk Factors for Hospital Admission

The patients included in this hospital-based study represent a sample population at high-risk for wheezing. Preschoool wheezing is one of the main causes of hospitalization in children aged <5 years [10], who suffer disproportionally more often from severe “asthma” exacerbations requiring emergency visits and hospital admissions compared to school-aged children [11]. Population-based data in French schoolchildren in the last year of nursery school (aged 5 years) indicated a lifetime prevalence of “asthma” of 11.0% [12]. In a birth cohort study conducted in Manitoba, Canada, the cumulative risk of hospitalization during the age interval 0–4 years for males and females were 2.1% vs. 1.1%, respectively [13]. Moreover, once admitted to hospital, the rate of emergency presentations (emergency department and readmission) within 12 months after discharge is high (20.5%) [11], and increases further with the number of prior hospitalizations and physician visits [13].

Preschool “asthma” and wheezing episodes are of multifactorial origin. Host (genetics, atopy) and environmental factors (viral and microbial exposure, exposure to passive smoking and indoor and outdoor air pollution, diet, exercise, vitamin intake, etc. [10] and low socioeconomic status [12] all play a role in asthma onset, prevalence and control. Moreover, gender may have a significant impact on asthma risk. In boys, the development of peripheral and bronchial airway dimensions relative to the growth of lung volume is delayed (mainly after the age of 4–5 years) vs. girls [14]. During the first year of life, males had approximately three times the probability of hospitalization due to asthma as females, and the cumulative risk of hospitalization during the preschool age for males was nearly twice that of females [13].

In our study, there were notable differences between the two groups (PTSE vs. no PTSE). For example, in the exposed group, male predominance is very marked. This confirms previous studies reporting higher hospitalization rates for asthma amongst boys than girls [6], and male sex as a risk factor for the recurrence of pre-school wheezing [15]. Prenatal tobacco smoke exposure is known to reduce birth weight [16]. Mean birth weight was lower in the exposed group, but the mean weights of the two groups were within the normal range, without any difference in the rate of intrauterine growth restriction. Regarding atopy and phenotypes predicting evolution to persistent asthma, the two groups are comparable, with atopy occurring in half of all hospitalized patients regardless of exposure. The mean age of hospitalization was between 8 and 11 months depending on the group. In the exposed group, the mean age of hospitalization and the age at first hospital admission were 2 months less in the exposed group, so a respiratory event is likely to have occurred earlier.

The declared rate of prenatal tobacco smoke exposure, greater than 33%, was higher than the 17% reported in France [17]. This may be related to a selection bias. Rates of other types of passive smoking were also different [18]. One-third of mothers reported smoking at least 10 cigarettes a day according to the literature [17], contrasting with one-quarter in the present study. Half of them reported a low level of consumption and most of them had smoked throughout their entire pregnancy. Most children who were exposed to prenatal tobacco smoke were also exposed subsequently to post-natal smoke. Of note, in a previous report, the odds ratio of maternal smoking for transient early wheezing (vs no wheezing) was 2.2 (95% CI 1.3–3.7) [7].

Taken together, the requirement for hospitalisation in the present study cohort may be due to the addition of many risk factors (PTSE+male sex+lower birth weight+ lower socioeconomic status). PTSE per se as a sole risk for acute wheezing may represent a health issue of lesser degree in the general population.

### 4.2. Clinical Impact and Economic/Social Consequences

The dose-effect impact of TPSE on hospitalization for wheezing which we have demonstrated cannot be compared directly with the literature, since studies analyzing the effect of prenatal tobacco smoke exposure until now have focused on the incidence and risk factors of wheezy dyspnea and asthma [19,20,21,22]. Presumably, more tobacco smoke exposes the foetus to greater structural changes and their clinical consequences. Liu et al. observed that low-intensity cigarette consumption during either the first or second trimester of pregnancy, even as low as 1–2 cig/day, was associated with an increased risk of preterm birth [23]. In a prospective analysis of a large birth cohort, Lanari et al. have reported that, when considering PTSE, having a mother smoking more than 15 cigarettes every day or a mother exposed to second hand smoke were associated with a significant risk increase of hospitalization for bronchiolitis of 3.5 (CI 1.5–8.1) and of 1.7 (1.1–2.6), respectively [24]. Carlsen et al. studied tidal breathing parameters at 3 days of life and demonstrated a significant reduction in change in tidal breathing expiratory flow to total expiratory time ratio (*t* PEF/*t* E) in infants exposed in utero to maternal smoking [25]. One daily cigarette corresponded to a change in *t* PEF/*t* E of −0.0021 ((95% CI) −0.0040 to −0.0002) and a change in Crs of −0.026 mL·cmH_2_O (95% CI −0.045 to −0.007 mL·cmH_2_O). The decrease was 0.023 and 0.29, respectively, in infants of an average smoker [25]. The ratios declined with increasing exposure [25]. All the above, together with our findings, suggest that there is a no-threshold dose-effect relationship between prenatal tobacco smoke exposure and the duration of hospitalization for wheezing in infants during the first 1000 days.

Apart from the economic impact of hospital admissions due to PTSE (discussed below), the real life relevance and treatment burden of acute wheezing episodes relate to more frequent visits to the hospital, longer hospital stays, and the hidden burden of long-term maintenance anti-asthma therapy. Parents of admitted children have generally been counseled regarding the risk of long-term COPD and lung cancer due to cigarette smoking, but most report total ignorance of the disruption to family life (including the care of other siblings) and the weariness during the first 1000 days of their child’s life, as a direct consequence of PTSE. The direct impact of PTSE includes: (1) parents’ loss of time and energy as a result of repeated long waiting times in the Accident and Emergency departments; (2) hospital admissions usually last for 2 to 4 days, and longer stays will mean more bronchodilator aerosols, oxygen therapy, oral steroids, aid for feeding difficulties, IV fluids, nosocomial infections (occasional re-admission of viral gastro-enteritis), antibiotics and investigations such as chest X-rays; (3) moreover, long-term maintenance inhaled therapies at home may be required (inhaled steroids and/or anti-leukotriene receptor antagonists-, since exposure to passive smoking decreases the efficacy of inhaled steroids [26]). In many cases, wheezing is due to a combination of smaller airways inherent to PTSE and viral episodes, which are not responsive to conventional anti-inflammatory drugs, but which lead to over-prescription by physicians, who are unaware of specific wheezing phenotypes.

### 4.3. Economic/Social Consequences

Overall, asthma-related costs vary significantly across countries, depending on several factors, in particular the type of health system available. The socio-economic cost of childhood asthma includes direct, indirect, and intangible costs [10]. Direct costs generally account for 50–80% of the total costs and include disease management (e.g., outpatient visits, visits to emergency services, hospital admission, medications), investigations and other costs (e.g., home care, transportation to medical visits and hospital) [10]. In our institution, each additional hospitalization day will cost between 614 and 118,800 € per child [27]. Indirect costs include school and/or work-related losses and early mortality [10]. A child with an asthma exacerbation loses on average 3–5 days school days and at least one of her/his caregivers loses the same working time. Intangible costs are unquantifiable (impairment of quality of life, limitation of physical activities and schooling and study performance), with consequent psychological effects such as depression and anxiety [10]. In essence, the overall social burden of asthma is considerable, not only for the child but also for his/her parents and carers. 

### 4.4. Pathophysiology

The pathophysiology of lung involvement due to antenatal smoking is now better known. Nictotine appears to be the main culprit. In vitro, in embryonic murine lung explants, nicotine stimulated lung branching and dysanaptic lung growth occur in a dose-dependent fashion [28]. This depends on the presence of alpha7 nicotinic acetylcholine receptors (nAChRs) [28] in pulmonary fibroblasts [29]. Nicotine interacts directly with alpha 7 nAChR to increase collagen accumulation in cartilaginous membranes and terminal airways as well as alveolar walls, therefore increasing airway wall thickness [29]. The greater inner and outer airway wall and the smooth muscle area and decreased alveolar attachment points (reduction of the surface complexity of the lung parenchyma; less elastic recoil) results in increased airway responsiveness i.e., excessive airway narrowing in response to irritants encountered during the postnatal period [30]. Other changes include epithelial cells proliferation, disruption of its cilia, and presence of inclusion bodies [31], upregulation of surfactant protein gene expression, induction of neuro-endocrine cell hyperplasia in fetal lungs, lowered serum IgG and decreased activity and numbers of natural killer cells [32]. On the maternal side, nutrition may be an important issue. Smokers during pregnancy had lower intakes of most micronutrients, in particular with respect to vitamin C and carotenoid, and this may impact lung development [33]. Antioxidant intake was lowest in young women who smoked [33]. Noteworthy, since much of the effect of prenatal smoking on offspring lung function is mediated by nicotine, it highly likely that e-cigarette use during pregnancy will have the same harmful effects as do conventional cigarettes [34].

### 4.5. Smoking Cessation

Prenatal smoking is arguably one of the most important modifiable risk behaviors for long-term child health. The data regarding the desire to receive smoking cessation support and information about the respiratory risks of PTSE underline the problem of prevention and support for smoking cessation: how effective they are, how they are implemented, and what do pregnant women gain from them. Furthermore, less than 10% of pregnant smokers report that their expired carbon monoxide was measured during their pregnancy and more than half expressed a desire for more appropriate smoking cessation support. Since passive smoking is a public health problem, several measures have been implemented in recent years to improve prevention. Other types of prevention and smoking cessation assistance have also been envisaged. Meta-analyses have evaluated the impact of specialized interventions within families regarding the effect of smoking cessation and the gain in children’s health. For example, one of the 18 trials analyzed showed a cessation rate of 23.1% in the intervention group versus 18.4% in the control group, so they can be considered to be worthwhile [35]. 

A suitable moment for learning is any circumstance during which a positive change in behavior can take place [36]. This might be a particular setting, offering an opportunity to promote awareness of the problem [37]. The motivation to quit smoking is all the greater if parents grasp the beneficial impact on their child’s health. Pregnancy is a key time for weaning and early management. The period during which a child is hospitalized is also a window of opportunity for identifying parents who smoke and offering them appropriate weaning assistance [38]. In addition to obstetricians, pediatricians can play a central role in this respect. They can greatly influence parents and put forward convincing arguments about smoking cessation and the positive effect this can have on their children. Another issue to be considered is the time and effort that pediatricians must devote to learn how to help parents to stop smoking. A randomized study evaluating the impact of a smoking cessation education program for pediatric pulmonologists and nurses in a Philadelphia children’s hospital reported significant results. The group that received specific training had a better approach to parents who smoked, provided higher level counselling and more practical help [39].

Efforts to promote prevention should be reinforced and probably include a minimum level of training for health workers for the evaluation of smoking, giving appropriate advice, and organizing dedicated meetings. Passive smoking has a significant cost in terms of pediatric morbidity and mortality. For example, in United States, it has been estimated that parental smoking is responsible for excess illness and death in children each year due to low birth weight (46,000 cases), unexpected infant deaths (2000 cases), RSV bronchiolitis (22,000 hospitalizations, 1100 deaths), and asthma (1.8 million outpatient visits and 14 deaths) [40].

Smoking reduction is more likely to occur than complete cessation during pregnancy and the message “less consumption entails less risk” seems too banal, given that even low cigarette consumption has negative repercussions on fetal development. A recent French study showed that even low exposure to prenatal tobacco smoke exposure was associated with a significant reduction in birth weight [17]. PTSE is an early and modifiable risk factor: it is the first on which all health professionals must mainly act.

### 4.6. Strengths

The strengths of the study were as follows. First, it is one of the few to attempt to quantify PTSE in a cohort of hospitalized infants. Second, the duration of hospitalization standardized to age improves the comparability of hospitalization times in infants hospitalized several months apart. Third, the long inclusion period (one and a half year) covered several seasons, thereby reducing the bias of seasonal epidemics. 

### 4.7. Limitations

First, this study was retrospective with a medium-sized cohort and limited power (more unexposed and low exposure groups than a high exposure group). The questionnaire participation rate was moderate (129/237 (54%)), and we did not perform any comparison of the demographic and clinical characteristics of participants vs. non-responded subjects. Second, a recall bias may have occurred [41,42]. Declarative reporting frequently results in underreporting of consumption due to guilt and underestimation of risk. Studies have shown that the reporting mode alone may underestimate the number of women smokers by 20% and the number of cigarettes smoked by 50% [20]. In addition, a memory bias was possible since we asked mothers about their cigarette consumption 1 to 2 years prior to the study. Third, the main result of the univariate analysis does not take into account other confounding factors that might increase the risk of preschool wheeze (place of residence, presence of mold in the child’s bedroom, ethnicity, breastfeeding, season, number of siblings, or community, fetal exposure to alcohol, drugs and environmental pollutants. Fourth, it is difficult to unravel the influence of the different types of passive smoking, which are often associated (98% of patients exposed to prenatal tobacco smoke exposure were also exposed to post-natal passive smoking). The measured impact probably arose from these two types of exposure. However, the multivariable analysis tended to offset this limitation.

## 5. Conclusions

We found a no-threshold dose-effect relationship between prenatal tobacco smoke exposure and the duration of hospitalization for wheezing in non-premature infants less than 2 years of age, indicating that there is no safe level or safe trimester regarding PTSE. Although future studies are required to confirm these findings, the medical community should act with more urgency by seeking out ways to provide more intensive and effective smoking cessation assistance to reduce the number of children exposed to passive smoking. Offering stronger smoking cessation education programs during pregnancy is a priority.

## Figures and Tables

**Figure 1 healthcare-09-01089-f001:**
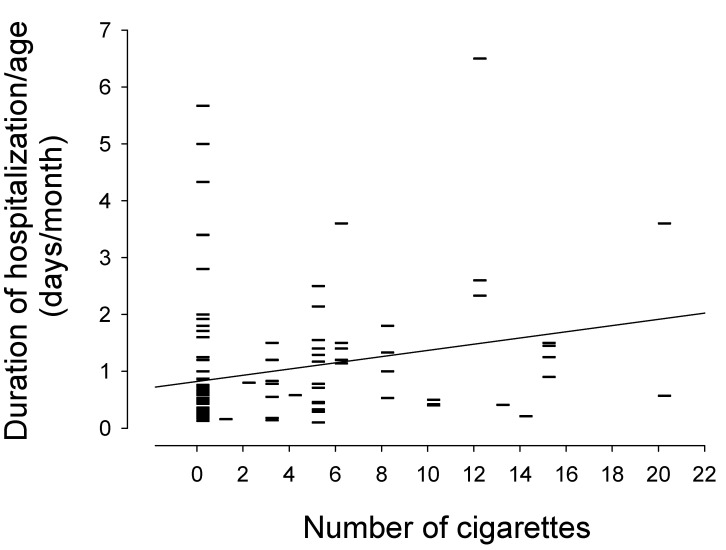
Linear correlation between duration of hospitalization/age and reported number of cigarettes smoked during pregnancy.

**Table 1 healthcare-09-01089-t001:** Demographic and General Characteristics of patients.

Characteristics		Total Cohort	Tobacco (PTSE) Unexposed	Tobacco (PTSE) Exposed	*p*-Value
**N**		129	82	47	
Sex	F	37 (29)	29 (35)	8 (17)	**0.03**
	M	92 (71)	53 (65)	39 (83)	
Birth weight (kg)		3.25 ± 0.43	3.35 ± 0.42	3.09 ± 0.39	**0.007**
IUGR		11 (9)	6 (7)	5 (11)	0.53
Familial atopy		68 (53)	44 (54)	24 (51)	0.85
Low social level		50 (39)	23 (28)	27 (57)	**0.001**
Personal atopy		59 (46)	38 (46)	21 (45)	1
Positive API		91 (71)	57 (70)	34 (72)	0.84
PIAMA Score ≥ 16		31 (24)	16 (20)	15 (32)	0.13
Mean age first hospitalization (months)		9.9 ± 6.38	10.9 ± 6.5	8.4 ± 5.9	**0.03**
Number of hospitalisations	1	67 (52)	44 (54)	23 (49)	
	>1	62 (48)	38 (46)	24 (51)	

*n* (%); Mean ± SD. API: Asthma Predictive Index; IUGR: Intra-Uterine Growth Retardation; PIAMA: Prevention and Incidence of Asthma and Mite allergy; PTSE: Prenatal Tobacco Smoke Exposure. Bold numbers indicate significant differences between the PTSE exposed vs. non-exposed patients

**Table 2 healthcare-09-01089-t002:** Comparison of included patients vs. all wheezing/asthmatic patients.

Variable	Included Patients	All Patients	*p*-Value
*n*	129	304	
Sex ratio (M:F)	2.3:1	2.3:1	
Age at First Wheezing Episode (mo)	10.0 ± 0.6	12.3 ± 0.4	0.0004
Number of Wheezing Episodes *	1.8 ± 0.1	1.2 ± 0.04	<0.0001
Cumulative duration (days) of Hospitalization *	8.7 ± 0.5	4.6 ± 0.2	<0.0001

*** Observed during the first 2 years of life.

**Table 3 healthcare-09-01089-t003:** Duration of the hospitalization/age (days/months) depending on prenatal exposure to tobacco.

Exposure	Median	Q1–Q3	Min–Max
Total cohort	0.6	0.35–1.2	[0.1–6.5]
Unexposed	0.58	0.3–0.86	[0.13–5.67]
Exposed (all groups)	0.9	0.44–1.5	[0.1–6.5]
Exposed [1–5] cig.	0.78	0.33–1.29	[0.1–2.5]
Exposed [6–10] cig.	1.13	0.5–1.4	[0.4–3.6]
Exposed [11–20] cig.	1.45	0.56–2.6	[0.21–6.5]

Cig: cigarettes per day; PTSE: Prenatal Tobacco Smoke Exposure; Q1–Q3: First and third quartiles.

**Table 4 healthcare-09-01089-t004:** Duration of hospitalization/age in multivariable analysis.

Variable	β	95% CI	Standardized β	95%CI Standardized β	*p*-Value
PTSE	0.06	0.01–0.11	0.27	0.03–0.51	0.03
Passive smoking pregnant women	0.29	−0.15–0.73	0.13	−0.07–0.33	0.20
Post-natal passive smoking	−0.34	−0.86–0.17	−0.16	−0.40–0.08	0.18
Gender (male)	0.05	−0.36–0.46	0.02	−0.15–0.20	0.81
Intra-Uterine Growth Restriction	0.69	0.04–1.35	0.03	0.01–0.35	0.04
Familial Atopy	−0.001	−0.37–0.36	−0.004	−0.17–0.17	0.96
Personal Atopy	0.11	−0.25–0.49	0.06	−0.11–0.23	0.53

β: regression coefficient; PTSE: Prenatal Tobacco Smoke Exposure.

**Table 5 healthcare-09-01089-t005:** Secondary endpoints.

Exposure Quantification	Dyspnoea Severity *	Hospitalization Experience Score **	Parental Work Absenteeism	Age First Wheezing Episode (Months)	Emergency Visits
**Unexposed**	53/82 (64.6)	2.45 ± 1.24 (1.82–3.07)	49/82 (59.8)	5 (2–8)	2 (1–4)
**Exposed (all groups)**	37/47 (78.7)	1.61 ± 1.40 (0.91–2.32)	26/47 (55.3)	4 (2–6)	2 (1–4)
**[1–5] cig./day**	17/23 (73.9)	1.82 ± 1.55 (1.04–2.06)	13/23 (56.5)	4 (3–6)	2 (1–2)
**[6–10] cig./day**	11/13 (84.6)	1.23 ± 1.23 (0.61–1.84)	8/13 (61.5)	2 (1–5)	3 (3–4)
**[11–20] cig./day**	9/11 (81.8)	1.63 ± 1.28 (0.99–2.27)	6/11 (54.3)	3 (2–5)	3 (1–5)

*n*/N (%); Mean ± SD (95% CI); median (Q1–Q3); Cig.: cigarettes; * Severe dyspnea was defined as oxygen requirement, and/or complications such as pneumonia or atelectasis; ** Mothers graded from 0 (very bad experience) to 5 (good) their experience of their child’s hospitalizations.

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
