# Peer review of "The First 1000 Days: Impact of Prenatal Tobacco Smoke Exposure on Hospitalization Due to Preschool Wheezing"

_healthcare, 2021, doi:10.3390/healthcare9081089_

Round 1

Reviewer 1 Report

This paper is a well written paper on a well designed observational study. 

I just have two questions: 

The study included patients only who had been hospitalized at least once for an episode of wheezy dyspnea. I think this is a high risk sample for wheezing. Here, 47 out of 129 had PTSE. These patients had longer hospitalization (.9 days/month). Is this result relevant if we consider how many children live in the hospital´s area or compared to the number of all children admitted to the hospital at the same time period? If 20% of all children have PTSE, then how many (%) of them were hospitalized for wheezing? I presume that most children with PTSE in the Bordeaux area were not hosptalized for wheezing, thus, PTSE as a cause for longer hospitalization (for wheezing) may be a smaller problem if your data of a small risk group are transferred to the population size? Please discuss.

Children with PTSE were slightly longer in hospital. This may be relevant or not. Are there data on what was the difference of remaining slightly longer in hospital? What happened in the 0.9 days long additional stay? Were there more interventions? More burdening assessment or treatment for the children? Please discuss the real life relevance (and/or additional burden to child) of the one additional day in hospital. 

Author Response

Reviewer 1

This paper is a well written paper on a well-designed observational study. 

We thank the reviewer for this kind comment

I just have two questions: 

Question 1

The study included patients only who had been hospitalized at least once for an episode of wheezy dyspnea. I think this is a high risk sample for wheezing. Here, 47 out of 129 had PTSE. These patients had longer hospitalization (.9 days/month). Is this result relevant if we consider how many children live in the hospital´s area or compared to the number of all children admitted to the hospital at the same time period? If 20% of all children have PTSE, then how many (%) of them were hospitalized for wheezing? I presume that most children with PTSE in the Bordeaux area were not hosptalized for wheezing, thus, PTSE as a cause for longer hospitalization (for wheezing) may be a smaller problem if your data of a small risk group are transferred to the population size? Please discuss.

Page 8 (para 2) and 9. These points are very important are have now been developed in section 4.1 :

Demographics and Risk Factors for Hospital Admission

The patients included in this hospital-based study represents a sample population at high-risk for wheezing. Preschoool wheezing is one of the main causes of hospitalization in children aged < 5 years {Ferrante, 2018 #813}, who suffer disproportionally more often from severe “asthma” exacerbations requiring emergency visits and hospital admissions compared to school-aged children {Donath, 2020 #817}. Population-based data in French schoolchildren in the last year of nursery school (age, 5 years) indicated a lifetime prevalence of “asthma” of 11.0% {Delmas, 2017 #814}. In a birth cohort study conducted in Manitoba Canada the cumulative risk of hospitalization during the age interval 0-4 years for males and females were 2.1% vs. 1.1%, respectively {Schaubel, 1996 #816}. Moreover, once admitted to hospital, the rate of emergency presentations (emergency department and readmission) within 12 months after discharge is high (20.5%) {Donath, 2020 #817}, and increases further with the number of prior hospitalizations and physician visits {Schaubel, 1996 #816}.

Preschool “asthma” and wheezing episodes are of multifactorial origin. Host (genetics, atopy) and environmental factors (viral and microbial exposure, exposure to passive smoking and indoor and outdoor air pollution, diet, exercise, vitamin intake, etc…) {Ferrante, 2018 #813} and low socioeconomic status {Delmas, 2017 #814} all play a role in asthma onset, prevalence and control. Moreover, gender may have a significant impact on asthma risk. In boys, the development of peripheral and bronchial airway dimensions relative to the growth of lung volume is delayed (mainly after the age of 4-5 years) vs. girls {Fayon, 2021 #826}. During the first year of life, males had approximately 3 times the probability of hospitalization due to asthma as females, and the cumulative risk of hospitalization during the preschool age for males was nearly twice that of females {Schaubel, 1996 #816}.

In our study, …/…

The declared rate of prenatal tobacco smoke exposure, …/…

Taken together, the requirement for hospitalisation in the present study cohort may be due to the addition of many risk factors (PTSE+male sex+lower birth weight+ lower socioeconomic status). PTSE per se as a sole risk for acute wheezing may represent a health issue of lesser degree in the general population.”

Question 2

Children with PTSE were slightly longer in hospital. This may be relevant or not. Are there data on what was the difference of remaining slightly longer in hospital? What happened in the 0.9 days long additional stay? Were there more interventions? More burdening assessment or treatment for the children? Please discuss the real life relevance (and/or additional burden to child) of the one additional day in hospital.

Page 10, Para  2. Thank you for giving us the opportunity to discuss this important matter. A new paragraph has been added as follows:

“Apart from the economic impact of hospital admissions due to PTSE (discussed below), the real life relevance and treatment burden of acute wheezing episodes relate to more frequent visits to the hospital, longer hospital stays, and the hidden burden of long-term maintenance anti-asthma therapy. Parents of admitted children have generally been counseled regarding the risk of long-term COPD and lung cancer due to cigarette smoking, but most report total ignorance of the disruption of family life (including the care of other siblings) and the weariness during the first 1000 days of their child’s life, as a direct consequence of PTSE. The direct impact of PTSE includes 1) Parents’ loss of time and energy as a result of repeated long waiting times in the Accident and Emergency departments. 2) Hospital admissions usually last for 2 to 4 days, and longer stays will mean more bronchodilator aerosols, oxygen therapy, oral steroids, aid for feeding difficulties, IV fluids, nosocomial infections (occasional re-admission of viral gastro-enteritis), antibiotics and investigations such as chest X-rays.  3) Moreover, long-term maintenance inhaled therapies t home may be required (inhaled steroids and/or anti-leukotriene receptor antagonists, since exposure to passive smoking decreases the efficacy of inhaled steroids [26]). In many cases, wheezing is due to a combination of smaller airways inherent to PTSE and viral episodes, which are not responsive to conventional anti-inflammatory drugs, but which lead to over-prescription by physicians who are unaware of specific wheezing phenotypes.”

Reviewer 2 Report

This manuscript reports a relationship between age-corrected hospitalizing days and prenatal smoking of mother in an infant population that experienced hospitalization due to wheezing dyspnea. The result found a positive association between the two suggesting some dose-dependent pathophysiological effects of maternal smoking on fetus/infant, which could have led to prolonged hospitalization when they developed wheezing dyspnea.

  1. The sample size is too small to obtain convincing result, though the authors self-recognize the size as “medium-sized”. Due to the small sample size, the univariate correlation result shown as Fig. 1 does not seem to be convincing, though maybe statistically significant. The relationship does not seem to be linear due to the great scattering of data points. At least the authors should not quantitatively discuss about the effect of smoking as e.g., 0.055 days/month/cigarette, based on this figure/correlation. If they still want to do, then they should clearly express uncertainty (e.g., 95% CI) to this value.
  2. Total of 237 subjects were included (means “recruited”?) and 129 of them responded. The participation rate cannot be regarded as “high” in contrast to the authors self-recognition. Moreover, there is no description about the differences/bias in demographic/clinical characters of responded and non-responded subjects.
  3. The authors are required to specify “multivariate analysis” they used. In this respect, Table 3 must be more detailed. They are also required how did they inspect the applicability of multivariate analysis to this data set, e.g., if they check normality of distribution, which is indispensable for applying some of the multivariate analyses.
  4. A study that uses age-corrected hospitalizing duration as an outcome may have merit from public health viewpoint as an impact of economic/social consequence of the disease but may have limited clinical implication because it is quite reasonable that hospitalizing duration does not exclusively correspond to severity of condition. However, the authors are not aware of this point. The manuscript should discuss more about the impact of prolonged hospitalization due to maternal smoking during pregnancy.

Author Response

Reviewer 2

This manuscript reports a relationship between age-corrected hospitalizing days and prenatal smoking of mother in an infant population that experienced hospitalization due to wheezing dyspnea. The result found a positive association between the two suggesting some dose-dependent pathophysiological effects of maternal smoking on fetus/infant, which could have led to prolonged hospitalization when they developed wheezing dyspnea.

  1. The sample size is too small to obtain convincing result, though the authors self-recognize the size as “medium-sized”. Due to the small sample size, the univariate correlation result shown as Fig. 1 does not seem to be convincing, though maybe statistically significant. The relationship does not seem to be linear due to the great scattering of data points. At least the authors should not quantitatively discuss about the effect of smoking as e.g., 0.055 days/month/cigarette, based on this figure/correlation. If they still want to do, then they should clearly express uncertainty (e.g., 95% CI) to this value.

Page 6, Figure 1. We confirm that the relationship is linear, although not very apparent in the previous version of the graph. We have changed the presentation so that the high numbers of babies not exposed to PTSE is more visible in the left lower corner (of the graph).

As suggested, 95% CI have been added to the manuscript, page 6, last para : ”In the univariate analysis, as depicted in Figure 1, there was a linear relationship between prenatal tobacco smoke exposure and duration of hospitalization/age (r = 0.238, slope = 0.055, [95% CI: 0,02-0,09], p = 0.006).” and to Table 3 (Page 7).

  1. Total of 237 subjects were included (means “recruited”?) and 129 of them responded. The participation rate cannot be regarded as “high” in contrast to the authors’ self-recognition. Moreover, there is no description about the differences/bias in demographic/clinical characters of responded and non-responded subjects.

We thank the reviewer for these comments.

Page 4, Para 2. Results section. The sentence has been changed as follows: “A total of 237 patients were recruited

Regarding the participation rate, we felt that a 50% rate was fairly high, but admittedly did not achieve the gold standard representative rate of > 80 %. And we did not perform any comparison of the demographic and clinical characteristics of participants vs. non-responded subjects. This has been modified in the manuscript.

Page 13. The following sentence has been deleted from para 4.5 Strengths: “Fourth, the questionnaire participation rate was rather high (54%).”

Page 13. The following has been added to para 4.6 Limitations: “First, this study was retrospective with a medium-sized cohort and limited power (more unexposed and low exposure groups than high exposure group). Also, the questionnaire participation rate was moderate (129/237 (54%)), and we did not perform any comparison of the demographic and clinical characteristics of participants vs. non-responded subjects.”

  1. The authors are required to specify “multivariate analysis” they used. In this respect, Table 3 must be more detailed. They are also required how did they inspect the applicability of multivariate analysis to this data set, e.g., if they check normality of distribution, which is indispensable for applying some of the multivariate analyses.

In response to this comment, we have modified the Staistical Analysis section, page 4, para 1: “We performed a univariate analysis and multivariable linear regression to search for an association between the duration our primary endpoint and PTSE, adjusted on potential confounders. The hypotheses of the model were verified: independence, linearity, normality and homoscedasticity.”

  1. A study that uses age-corrected hospitalizing duration as an outcome may have merit from public health viewpoint as an impact of economic/social consequence of the disease but may have limited clinical implication because it is quite reasonable that hospitalizing duration does not exclusively correspond to severity of condition. However, the authors are not aware of this point. The manuscript should discuss more about the impact of prolonged hospitalization due to maternal smoking during pregnancy.

In response to a similar comment by the co-reviewer, a new paragraph has been added as follows: Page 10, Para  2 (Please see Question 2 above)

Page 11, para 1; A new paragraph has been added:

“Overall, asthma-related costs vary significantly across countries, depending on several factors, in particular the type of health system available. The socio-economic cost of childhood asthma include direct, indirect, and intangible costs [10]. Direct costs generally account for 50–80% of the total costs and include disease management (e.g., outpatient visits, visits to emergency services, hospital admission, medications), investigations and other costs (e.g., home care, transportation to medical visits and hospital) [10]. In our institution, each additional hospitalization day will cost at least 1 188,00 € per child [27]. Indirect costs include school and/or work-related losses and early mortality [10]. A child with an asthma exacerbation loses on average 3–5 days school days and at least one of her/his caregivers loses the same working time. Intangible costs are unquantifiable (impairment of quality of life, limitation of physical activities, and schooling and study performance), with consequent psychological effects such as depression and anxiety [10]. In essence, the overall social burden of asthma is considerable, not only for the child but also for his/her parents and carers.“

Round 2

Reviewer 2 Report

The authors responded to most of my comments, but the followings require some more response.

  1. The authors are advised to carry out statistical comparison between responders and non-responders of this study.
  2. Specify the multivariate analysis: Is it multiple linear regression analysis? Specify the program of SAS used for the analysis in Statistical Method section.
  3. Was "beta" in Table 3 is, then, unstandardized partial regression coefficient? The CI in this table should be on the row next to beta. The CI for passive smoking seems not correct if this is CI for beta 0.0036. Check it. Standardized partial regression coefficient should also be added to this table so that the relative contributions of each variables can be compared by the readers.      

Author Response

Point by point Correction: Healthcare-1282711: REVIEWER 2

The first 1000 days: Impact of prenatal tobacco smoke exposure on hospitalization due to preschool wheezing by Cyrielle Collet, et al.

  1. The authors are advised to carry out statistical comparison between responders and non-responders of this study.

Page 4, last para. As suggested, we have added the following to the manuscript: “Compared to the population of patients with wheezing/asthma admitted during the same time frame, included patients were younger upon their first wheezing episodes and had more severe disease. However, the sex ratio was identical, with a greater proportion of males (Table 2).“

Page 5. A new Table (2) has been added

  1. Specify the multivariate analysis: Is it multiple linear regression analysis? Specify the program of SAS used for the analysis in Statistical Method section.

Page 4, Para 1. Statistical Analysis. The following has been added: “A linear multivariable regression was performed with Sas 9.4 using Proc reg. Standardized coefficients were calculated, and the same procedure applied to them.”

  1. Was "beta" in Table 3 is, then, unstandardized partial regression coefficient? The CI in this table should be on the row next to beta. The CI for passive smoking seems not correct if this is CI for beta 0.0036. Check it. Standardized partial regression coefficient should also be added to this table so that the relative contributions of each variables can be compared by the readers.

Page 7. Table 3 (now table 4) has been modified, to add standardized partial regression coefficients, and checking/correcting the results, as suggested.

We would like to thank the reviewer for this very important comment, and valuable addition to the manuscript. The added analysis is equivalent to centre-reducing all the variables, and therefore to finally "erase" the unity of each variable. This has allowed us to compare the effect of each variable to the others, and we show that the effect of smoking during pregnancy has the greatest impact on the standard deviation of the length of hospitalization.
